# Modification of Starches and Flours by Acetylation and Its Dual Modifications: A Review of Impact on Physicochemical Properties and Their Applications

**DOI:** 10.3390/polym15142990

**Published:** 2023-07-09

**Authors:** Edy Subroto, Yana Cahyana, Rossi Indiarto, Tiara Aray Rahmah

**Affiliations:** Department of Food Industrial Technology, Faculty of Agro-Industrial Technology, Universitas Padjadjaran, Bandung 45363, Indonesia; y.cahyana@unpad.ac.id (Y.C.); rossi.indiarto@unpad.ac.id (R.I.); tiara16005@mail.unpad.ac.id (T.A.R.)

**Keywords:** acetylation, starch, flour, physicochemical properties, modification

## Abstract

Various modification treatments have been carried out to improve the physicochemical and functional properties of various types of starch and flour. Modification by acetylation has been widely used to improve the quality and stability of starch. This review describes the effects of acetylation modification and its dual modifications on the physicochemical properties of starch/flour and their applications. Acetylation can increase swelling power, swelling volume, water/oil absorption capacity, and retrogradation stability. The dual modification of acetylation with cross-linking or hydrothermal treatment can improve the thermal stability of starch/flour. However, the results of the modifications may vary depending on the type of starch, reagents, and processing methods. Acetylated starch can be used as an encapsulant for nanoparticles, biofilms, adhesives, fat replacers, and other products with better paste stability and clarity. A comparison of various characteristics of acetylated starches and their dual modifications is expected to be a reference for developing and applying acetylated starches/flours in various fields and products.

## 1. Introduction

Native starches/flours generally have several drawbacks related to functional, pasting, and physicochemical properties, which can limit their use in various applications. These limitations include low swelling ability, absorption capacity, solubility, starch clarity, and freeze stability [1,2,3,4]. Most of the starches tend to retrograde easily when the starch paste is stored at low temperatures. This is caused by the amylose chains that had previously come out of the granules binding to each other again to form a crystalline structure [5,6]. Retrogradation causes an increase in starch viscosity, crystallinity, gel structure, and gel texture [7,8]. Various treatments and modifications have been used to improve these properties, one of which is by modifying acetylation. Acetylation is a modification involving the substitution of hydroxyl groups with acetyl groups; the number of substituted acetyl groups affects the characteristics of starch/flour [9,10,11].

The modification of acetylation in starch/flour has been reported to increase swelling ability, clarity of starch paste, and stability of starch against retrogradation [12,13,14]. Acetylated starch is often applied to improve the texture and appearance of products whose quality may decrease due to damage during processing or retrogradation. Acetylated starch can also provide a good thickening effect in various foods. However, acetylated starch is unstable to thermal processes characterized by increased breakdown viscosity [15,16,17]. This can be overcome by combining acetylation with other modifications that can improve thermal stability, such as cross-linking modifications and hydrothermal treatments, such as heat moisture treatment (HMT) and annealing (ANN). Hydrothermal modification can improve the formation of amylose-lipid complexes and the regularity of the crystalline matrix to control the swelling capacity and increase stability to heating and friction [18,19,20].

Modifying acetylation combined with cross-linking or hydrothermal treatment can improve the clarity of the paste, texture, and thermal stability [21,22]. Several studies have stated that hydrothermal treatment is able to increase the effectiveness of acetylation reactions with starch molecules so that more acetyl groups are substituted and can minimize the use of chemicals for the acetylation process [10,23]. However, several dual-modified treatments have contradictory effects, so the resulting characteristics depend on the dominant treatment [10,22].

Changes in the characteristics of starch/flour due to the modification of acetylation are highly dependent on the degree of substitution (DS) and treatment conditions such as the source of starch, type of reagent, pH, temperature, and time [12,15,24]. The starch source determines the amylose-amylopectin content, which determines the amorphous and crystalline structures of the starch granules. This type of acetylation reagent generally uses acetic acid, vinyl acetate, or acetic anhydride, which can be catalyzed using bases such as NaOH and KOH. At the same time, temperature and time reaction determine the level of DS obtained, which greatly affects the characteristics of the starch/flour produced. This review describes studies on modifications of various types of starch and flour by acetylation or dual modification of their physicochemical characteristics, as well as their applications in various fields/products, so that they can become a reference for the development of starches/flours.

## 2. Applications of Acetylated Modified Starch/Flour

Modifying a starch affects its characteristics, and the changes that occur depend on the type of modification applied. Chemical modifications such as acetylation weaken the starch’s structure, thereby increasing its hydration capacity and reducing its tendency to retrograde [25,26]. Meanwhile, the combination of acetylation with cross-linking and hydrothermal modification (HMT and ANN) can increase the orderliness of the crystalline matrix so that the gelatinization process becomes slower and granule swelling is limited [1,27,28]. Therefore, acetylated modified starch or its dual modification is often used to improve the hydration quality and thermal stability of the product [24,29]. However, the application of acetylated starch and its dual modifications has been developed in various fields and various products. Some applications of acetylated modified starches/flours for various products and fields can be seen in Table 1.

Acetylation modification can increase the functionality of starch and its applications, especially in foods. The high hydration ability of acetylated starch has the potential to be used as a thickening agent. Several studies also reported that acetylation modification has good stability and resistance to retrogradation and syneresis, so it has the potential to be applied as a stabilizer in products that require low-temperature storage [13,45]. In addition, acetylation modification can increase OAC so that it can be applied as a filming agent. Acetylated starch has many applications in the food industry, some of which are in products such as retorted soups, sauces, canned pie fillings, frozen food, baby food, and salad dressings [15,24,46].

Acetylated modified starch, especially in nanocrystal form, has been effectively applied for encapsulation and as a delivery system for various drugs and other active compounds [47]. de Oliveira et al. [30] applied acetylated cassava starch as starch-based nanoparticles for the encapsulation of antioxidants; it was reported that acetylated cassava starch interacted well with antioxidant compounds, especially BHT and protected antioxidants from the degradation process, and increased the thermal stability of nanoparticles. Liu et al. [32] applied acetylated debranched waxy corn starch as a nanocarrier for curcumin, and it was reported that curcumin micelles of acetylated starch had a spherical shape with a particle size of about 50–100 nm and could accommodate curcumin until the concentration of 0.45 mg/mL. Gangopadhyay et al. [33] applied retrograded acetylated corn starch to drug (budesonide) delivery, and it was reported that tablets from retrograded acetylated corn starch were able to release the drug in ileocolonic by 81.38% and were potentially suitable for the treatment of ileocolonic diseases. Meanwhile, Xiao et al. [31] reported that acetylated rice starch nanocrystals could be used for protein (BSA) delivery by significantly slowing BSA protein release.

Acetylated modified starch, especially in nanocrystal form, has been applied to improve the quality of bioactive films. Promhuad et al. [34] applied acetylated cassava starch and Maltol Incorporated in active biodegradable film/packaging fabrication; it was reported that film based on acetylated cassava starch, which was incorporated with 10% maltol, reduced molecular mobility, hydrophilicity, elongation, and tensile strength. The active film based on acetylated cassava starch inhibited the fungal growth by up to six times longer and maintained the flavor of bakery products. Fitch-Vargas et al. [35] applied acetylated corn starch in the manufacture of starch-based bioplastics; it was reported that acetylated corn starch improved homogeneity and mechanical properties, while the solubility of starch-based bioplastics decreased to 24.9–28.2%. Meanwhile, Nasseri et al. [36] reported that acetylated corn starch could be applied to biodegradable polymers poly(lactic acid) for packaging materials, where acetylated corn starch can increase the thermal stability of biodegradable polymers and improve mechanical properties such as toughness and tensile strength.

Acetylated modified starch and its dual modification can also be used as a stabilizer in various emulsion products and can act as a fat replacer. Yao et al. [37] applied acetylated cassava starch nanoparticles as an emulsion stabilizer; it was reported that acetylated starch nanoparticles improved hydrophobicity and improved emulsion capacity by improving the droplet size and homogeneity so that the storage stability increased up to 35 days. Cui et al. [38] applied cross-linked acetylated cassava starch to the manufacture of set yogurt, and it was reported that cross-linked acetylated cassava starch improved the stability, viscous modulus, and elastic modulus of the set yogurt. Meanwhile, Osman et al. [16] utilized acetylated corn starch as a fat replacer in beef patties, and it was found that it was suitable as a fat replacer for meat products. At the same time, the use of 15% acetylated corn starch improved the acceptance of organoleptic, microstructural, and physicochemical properties in beef patties.

Acetylated starch can also be applied directly in the manufacture of various food products such as bread and noodles. For some food products, such as bread, retrogradation causes bread stalling, where the quality of the bread decreases in the form of a harder texture [48]. Acetylated starch could slow down and improve retrogradation, thereby improving the texture and quality of bread. Rahim et al. [39] applied acetylated arenga starches to bread making; it was reported that adding acetylated starch up to 50% was able to improve the quality of the bread produced, which included sensorial properties, oven spring, oil absorption, and oil holding capacity. Meanwhile, Lin et al. [40] applied acetylated corn starch to the manufacture of noodles, and it was reported that acetylated corn starch increased the brightness and reduced the tensile properties, chewiness, adhesion, and hardness of noodles. Acetylated starch was also reported to increase resistant starch and slowly digestible starch in noodles. Wang et al. [41] also reported that acetylated rice starch and potato starch improved the gut microbiota fermentation by producing more SCFA and were easier to use and more quickly fermentable by the gut microbiota.

Acetylated modified starch and its dual modification can also be used in the chemical field, namely as a waste treatment coagulant and as an adhesive, especially to increase its water resistance and shear strength [49]. Gu et al. [42] applied acetylated-crosslinked corn starch as a wood-based panel adhesive, where the adhesive had better water resistance up to 1 MPa, and the adhesive was also heat-resistant so that it could be used in high-temperature pressing. Wang et al. [43] applied acetylated waxy corn starch as a wood adhesive, and it was found that the adhesive’s resistance to water increased up to 61.1%, and the shear strength increased up to 321% in the wet state and 59.4% in the dry state. Meanwhile, Posada-Velez et al. [44] applied acetylated potato starch and corn starch as coagulants for wastewater treatment, and it was found that acetylated starch from both corn and potato starch had good effectiveness as a coagulant for wastewater treatment by significantly reducing pH, color, turbidity, and electrical conductivity.

## 3. Acetylation Modification Process in Starch/Flour

### 3.1. Mechanism of Acetylation

The characteristics of starches and flours could be improved by chemical, physical, and combination or dual modifications. These modifications aim to change some of the properties of starch by altering its original structure through physical treatment or changing the hydroxyl groups in starch through chemical reactions such as oxidation and esterification [50,51,52,53]. The modification of starch includes the use of heat, oxidizing agents, alkalis, acids, and other chemicals that will generate new chemical groups, resulting in changes in size, morphology, molecular structure, and other physicochemical characteristics [54,55].

Chemical modification can change the significant characteristics of starch and flour. Chemical modifications can be carried out through acid hydrolysis, oxidation, cross-linking, and the addition of functional groups such as acetylation. In general, chemical modification adds new functional groups to the starch, which then affect the physicochemical properties of the modified starch [52,56,57,58]. The chemical modification of acetylation has been widely applied to various food industries. Several studies have been developed by combining acetylation modification with physical or other chemical modifications, such as acetylation + hydrothermal and acetylation + crosslinking, in order to increase starch’s functional value and expand its application [26,59,60].

Acetylation is a chemical modification technique conducted through the esterification of starch using acetic anhydride, acetic acid, and vinyl acetate reagents and alkali (NaOH, KOH, Ca(OH)_2_, and Na_2_CO_3_ as catalysts [61,62]. The basic principle of the acetylation reaction is the substitution of starch-free hydroxyl groups with acetyl groups (Figure 1) by weakening the bonds between starch molecules to produce starch that is amphiphilic (hydrophilic and hydrophobic) [63,64]. Acetylation is an indirect esterification process, so it is necessary to add a catalyst so that the reaction can take place. Before the reaction, the starch is first conditioned in an alkaline state to form the starch base. An acetate reagent is then added to form starch acetate [30,65,66,67]. The basic principle of the acetylation reaction by the substitution of starch-free hydroxyl groups with acetyl groups can be seen in Figure 1.

Figure 1 shows that the acetylation reaction occurs via the substitution of the acyl group with the free hydroxyl group portion of the glucose monomers as a constituent of starch molecules. There are many hydroxyl groups, so more free hydroxyl groups being substituted with acetyl groups will result in a greater degree of substitution. Acetylation can reach equilibrium, and if the reverse reaction occurs, it indicates the hydrolysis reaction of the ester bond. The rate of the acetylation reaction is affected by several factors as well as other esterification reactions, especially the structures of the acids and alcohols that react and the catalyst used [68]. The catalyst usually uses strong bases such as KOH and NaOH using anhydride reagents such as acetic anhydride. However, using organic acids or carboxylic acids as organocatalytics is also competitive to produce safer and more environmentally friendly processes. Organocatalytics that can be used include L-aspartic, citric, L-tartaric, L-malic, L-lactic, glycolic, fumaric acids, and L-proline. These organic catalysts can also produce high esterification reaction rates and high degrees of substitution [69,70].

The degrees of substitution (DS) affect the physicochemical and functional characteristics of acetylated modified starch/flour [71,72]. The DS values ranged from 0.01 to 3, describing the number of substituted acetyl groups in one glucose unit. Starch acetate with DS 0.01 indicates that there is one substituted acetyl group in 100 units of glucose. In contrast, starch acetate with DS 3 indicates that there are 300 substituted acetyl groups in 100 units of glucose. This is based on the theory that acetylation reactions can substitute three free hydroxyl groups of glucose units on C2, C3, and C6 atoms with acetyl groups [16,73]. Acetic starch with low DS has been widely applied in the food industry for many years. The Food and Drug Administration (FDA) stipulates that the maximum permissible limit for the percentage of acetyl groups in foodstuffs is 2.5% or the equivalent of a DS value of 0.1, but generally, for commercial food products, starch acetate is used with a low DS (<0.1) or medium DS (0.1–1.0) [15,74].

### 3.2. Effect of Acetylation Methods on Properties of Starch/Flour

The acetylation method involving reactant types and concentrations can affect the acetylation reaction’s efficiency. The DS of starch acetate is greater when the reactant concentration is high [75]. Generally, the ability and efficiency of acetic anhydride in substituting acetyl groups were greater than that of vinyl acetate at the same conditions and concentrations. In addition, the type of catalyst and reaction medium (solvent) also affect the efficiency of the reaction. Solvents that can be used in acetylation reactions include water, pyridine, and DMSO. Pyridine and DMSO have greater efficiency than water but can have adverse environmental and health impacts [24]. Thus, the number of substituted acetyl groups is affected by several factors, including the source or type of starch, the concentration of the reactants, pH, reaction time, and the catalyst used [65,76]. The starch type with many amorphous parts, such as starch from tubers, is more easily penetrated or substituted by acetyl groups. An example is the acetylation of potato starch, which produced a higher DS (DS: 0.180–0.238) than corn starch (DS: 0.133–0.184) [77]. The use of reactants with higher concentrations can produce higher DS; for example, the use of an acetylation of sweet potato starch using acetic anhydride at concentrations of 2, 4, 6, and 8% produced DS of 0.032, 0.059, 0.091, and 0.123 respectively [78]. pH can also affect DS, where the use of a higher pH could increase DS; for example, the acetylation of yellow pea starch at pH 9–10 produced a higher DS (DS: 0.071) than pH 7.5–9.0 (DS: 0.066) [62]. Longer reaction times can increase DS; for example, a chestnut starch acetylation at 30, 60, and 90 min reaction times resulted in the DS being 0.010, 0.020, and 0.024, respectively [14]. The type of catalyst used can also affect the DS; for example, the use of catalysts of Ca(OH)_2_, NaOH, and KOH 1 N in the acetylation of waxy cornstarch produced DS of 0.077, 0.081, and 0.085, respectively [61]. The DS value can be determined by several techniques, including headspace gas chromatography (HS-GC), infrared spectroscopy or FT-IR, nuclear magnetic resonance (NMR), and titration [53]. Several methods of modifying acetylation on various types of starch/flour and their effect on DS values can be seen in Table 2.

Infrared spectroscopy or FT-IR is also frequently used to determine the DS value of acetylated modified starch/flour. An example of the effect of acetylation modification on the FTIR spectra of starch can be seen in Figure 2. The FTIR spectrum of acetylated starch is shown by the appearance of several peaks or new absorption bands at wave numbers 1240, 1375, 1435, and 1754 or 1742 cm^−1^, which interprets C–O carbonyl stretching vibrations, CH_3_ symmetry deformation vibrations, CH_3_ antisymmetric deformation vibrations, and carbonyl C=O, respectively [98,99,100]. The appearance of these new absorption peaks indicates the occurrence of an esterification reaction, namely the substitution of hydroxyl groups by acetyl groups. The greater intensity of the absorption peaks indicates that the DS value is also greater and is followed by a weakening of the peaks at wave numbers 3421, 1082, and 1014 cm^−1^, which interprets the reduction of hydroxyl groups. Additionally, the spectrum of acetylated starch also shows that the anhydroglucose units tend to shift towards high wave numbers [100].

The DS value of acetylated starch can also be determined based on nuclear magnetic resonance (NMR) spectra, which generally use ^1^H NMR, ^13^C NMR, or ^13^C–1H COSY. NMR spectra can show changes in anhydroglucose units due to the substitution of hydroxyl groups by acetyl groups. An example of the effect of acetylation modification on the NMR spectra of starch can be seen in Figure 3.

The hydroxyl (-OH) groups of C_2_, C_3_, and C_6_ show a proton signal from the anhydroglucose unit between 4.4 and 5.6 ppm; the signal at 5.10 ppm corresponds to the anomeric proton of the link (α-1, 4), whereas the anomeric proton of the junction (α-1,6) exhibits a small signal at 4.86 ppm [101,102]. Figure 3 shows that in acetylated starch, an addition signal between 1.8 and 2.2 ppm indicates a methyl proton from the acyl group, thus indicating that the acetylation process was successful. The level of acetylation increases with the length of the reaction time. The spectrum of native starch shows four main signals (Signals 2, 3, 4, and 5). In comparison, in acetylated pea starch, there are two additional signals, namely signal 1 at 16.4 ppm and signal 6 at 166.6 ppm, which interpret the carbon of the methyl proton from the acyl group (-CH_3_) and the ester group (C=O). The longer the reaction time, the greater the area of the two signals, which means that the DS value of the acetylation increases [101].

Some of the advantages of acetylation modification include increasing starch clarity, lowering the gelatinization temperature, increasing the stability of frozen storage, and being more resistant to retrogradation [24,103]. Therefore, acetylated starch is commonly used in the food industry for the production of salad dressings, retorted soups, frozen foods, baby food, and snacks [86,104]. Chang and Lv [105] reported that acetylation reactions could also produce resistant starch in the form of RS type 4, which has a low glycemic index. In addition to its application in food ingredients, several studies have stated that acetylated starch with high DS is commonly used as a packaging material or in cigarette filters and has several applications in the pharmaceutical field [96,106]. Although it can improve the characteristics of starch, this modification tends to produce starch that is less resistant to high-temperature heating. Therefore, several researchers carried out a combination of physical modification + acetylation to improve the thermal stability of starch. Several modification combinations have been carried out, including annealing-acetylation [10,97] and sonication + acetylation [63].

## 4. Characteristics of Acetylated Modified Starch/Flour

### 4.1. Functional Properties of Acetylated Modified Starch/Flour

Functional characteristics are the properties of starch/flour that affect the usability of starch when applied. The functional characteristics of starch/flour can be viewed through several parameters, including amylose leaching/solubility, oil absorption capacity (OAC), water absorption capacity (WAC), freeze-thaw stability (FTS), and swelling power (SP). Swelling power is related to the amount of water absorbed, where the greater the swelling power value, the more water is absorbed. Starch granules will swell when heated in water, which is caused by the breaking of hydrogen bonds between starch molecules so that starch molecules will bind with water [54,107].

The swelling of granules is closely related to the release of starch molecules from granules or what is known as amylose leaching. As gelatinization proceeds, the water present in the starch suspension enters the outer amorphous and crystalline regions (located near the amorphous lamellae) [108,109]. The process of entry of water into the amorphous region causes granule swelling and weakens the hydrogen bonds between amylose and amylopectin chains. The continuous heating process takes place, causing starch molecules dissolved in water to easily move in and out of the solution system. Starch molecules that dissolve in hot water (amylose) will come out with the water, causing amylose leaching [110,111]. Solubility and swelling power are affected by several factors, including the amylose-lipid complex, molecular weight, granule size, amylose/amylopectin ratio, distribution of amylopectin chain lengths, amylose-amylopectin chain interactions, the molecular structure of starch granules, and crystal arrangement [28,112,113,114,115]. The solubility is more affected by the amylose content, while the swelling power is affected by the amylopectin component [116].

Water absorption capacity (WAC) is defined as the ability of granules to absorb water. WAC determines the amount of water available for starch gelatinization during cooking [117]. The gel formation cannot reach optimum conditions if the amount of water is lower. The greater the WAC, the more starch constituent material is lost, while the lower the WAC, the more compact the structure. Solubility, swelling power, and WAC are related to other parameters, namely viscosity, and crystallinity. Increases in solubility, SP, and WAC are associated with decreased crystallinity of starch and cause an increase in viscosity [118,119].

Starch paste stored at freezing temperatures will cause the water contained in the paste to form ice crystals, and these ice crystals will melt during thawing and release of water from the granules, which is called syneresis [120]. Freeze-thaw stability (FTS) describes the ability of pasta to withstand repeated freezing and thawing cycles without any physical changes. FTS is correlated with retrograde tendencies. The increased aggregation of starch molecules causes an increase in the release of water molecules from starch granules, so the syneresis increases, and FTS decreases [6,89]. Modification treatments, both acetylation and hydrothermal, can affect the interactions between starch molecules, which can affect the stability of the paste in frozen storage [74,121].

Retrogradation occurs when starch components that have been gelatinized re-associate. This association causes the starch structure to become more compact, making the paste more turbid. The modification of acetylation in corn starch was reported to increase starch’s stability against retrogradation and increase the clarity of starch paste [15,87]. Based on the level of clarity, the paste is classified into three types, namely transparent, moderately transparent, and cloudy. The clarity of the starch paste can be observed through a spectrophotometer by measuring the % transmittance, where the higher the %transmittance, the more transparent the paste. The level of clarity of the paste varies depending on the type of paste, solubility, swelling power (SP), and the aggregation of starch molecules [122,123]. Starch with high clarity and viscosity is suitable for application as a thickening agent, while some food products, such as salad dressings, require opaque starch [9,15,124].

Starch with high solubility and hydration ability is needed in several processed food industries, such as in the manufacture of noodles, bakery products, jelly, and many more [56,125]. Acetylation modification can increase SP and WAC, while hydrothermal modification generally causes a decrease in SP and WAC. Information regarding the effect of acetylation modification and its dual modification on the functional properties of starch/flour is available in Table 3.

Modification by acetylation could increase the swelling power, solubility, and WAC as the DS increases. The modification of acetylation causes an increase in SP in sweet potato starch [128], potato starch [10,77,83], purple yam, white yam, and cocoyam starches [84,85], sword bean starch [90], chickpea and yellow pea starches [62], corn and waxy corn starches [10,77,87], rice and waxy rice starches [10,88], sago starch [93], wheat starch [92], and waxy barley starch [10]. This increase is due to the substitution of the hydrophilic (acetyl) group, which facilitates water penetration into the starch granules [87,89].

Improvements in starch hydration and water absorption capacity after modification by acetylation were found in sweet potato starch [128], potato starch [83], purple yam, white yam, and cocoyam starches [84,85], sword bean starches [90], oat starch [89], rice starch [88], and sago starch [93]. The increased hydration power of the granules due to the substitution of acetyl groups causes increased flexibility of the structure, making it easier to bind water [46]. Increased starch solubility due to modification of acetylation was found in sweet potato starch [128], potato starch [77,83], sword bean starch [90], corn and waxy corn starch [77,87], rice starch [88], and sago starch [93].

The modification of acetylation can also affect the solubility of starch and flour. Acetylation will generally increase solubility, mainly due to the addition of acetyl groups that disrupt the interactions between starch molecules, which in turn increases the affinity of starch molecules to dissolve in water [88]. The increase in starch solubility due to the substitution of acetyl groups will weaken the hydrogen bonds that connect starch on an intermolecular level as well as inhibit intermolecular associations so that the starch molecules will bind with water and dissolve along with the water coming out of the starch granules [84]. Nevertheless, several studies have found a decrease in WAC in acetylated potato and corn starch [77], a decrease in solubility in acetylated cocoyam starch [85], and a decrease in SP in acetylated japonica and indica rice starch [91]. The effect of acetylation modification on the hydration and solubility of granules is influenced by the number of substituted acetyl groups (DS); the higher the DS, the greater the changes that occur [88,93].

Several studies reported that the combination of modification by esterification (acetylation) + hydrothermal treatment was able to increase SP, WAC, and starch solubility [10,23,97]. This is because the esterification modification is related to the substitution of ester groups (acetyl/hydroxypropyl), which weakens the granule structure, thereby facilitating the penetration of water into the granules and increasing the interaction of starch molecules with water [10,23].

The acetylation modification can also reduce retrogradation tendencies and increase the clarity level of starch paste [46,59,77,85,128,129,131]. Increasing the clarity of the paste is related to the hydration ability of the granules. The greater the hydration ability of the granules, the higher the level of clarity. The substitution of acetyl groups will limit the intermolecular interactions of starch, thereby reducing the tendency for starch syneresis [87,89,129]. The decrease in syneresis tendencies in acetylated starch is related to the substitution of acetyl groups, which limits intermolecular interactions when the paste is stored at low temperatures [46]. In addition, the modification of acetylation and its combination can also affect the whiteness of starch, where most of the modification treatments cause a decrease in whiteness [10,26].

Modification by acetylation can lead to an increase in the hydration ability of starch granules. This is due to the disorganization of the intragranular structure followed by the following events: (1) disruption of the intermolecular hydrogen bonds of the starch and increased penetration of water into the amorphous regions [75,88], (2) the presence of repulsive forces rejecting intermolecular starch due to the substitution of acetyl groups, (3) the partial depolymerization of the amylopectin structure, which causes a decrease in molecular weight (MW) [88,132], (4) a decrease in starch crystallinity [92,132,133] and the limitation of starch intermolecular interactions [84].

The acetyl group can have two properties, namely hydrophilic and hydrophobic. Acetylation at low DS makes starch tend to be hydrophilic, while acetylation at high DS makes starch tend to be hydrophobic [93,134]. Therefore, several studies reported that acetylation causes an increase in OAC [85,89,90,93]. The high OAC value has the potential to improve the flavor and mouthfeel of food products, such as whipped cream, sausages, chiffon cakes, and various processed desserts [135].

Several studies have reported that acetylation can reduce the tendency of starch to experience retrogradation and syneresis and shows a better level of clarity and transparency of starch than natural starch [77,85,89]. This is due to the substitution of acetyl groups which will hinder the association between starch molecules [88]. The clarity of starch is related to the ability of starch hydration (WAC and SP). The increased hydration ability of the granules causes more water molecules to be trapped in the granules so that when observed using a spectrophotometer, more light can be transmitted by the paste [136].

Acetylation can reduce the occurrence of syneresis in starch. A decrease in the percentage of syneresis indicates an increase in the stability of the paste against frozen storage [13,15]. Syneresis is related to retrogradation events in which reassociation occurs between starch molecules when stored at low temperatures. Syneresis causes a decrease in the water content of amylose/amylopectin, which can affect product characteristics. The substitution of acetyl groups can prevent the reassociation of the starch molecules so that the retrogradation tendency decreases. In addition, the substitution of acetyl groups also increases the water storage capacity so that the retention power of granules to water is also greater [10,46,83,88,130]. Mendoza et al. [84] reported that limiting the intermolecular associations of starch due to acetylation was necessary to produce starch with high hydration, good storage stability, and high clarity.

Acetylation modifications can also be combined with other modifications. The combination of acetylation with hydrothermal treatments (ANN and HMT) can increase the hydration and solubility of starch [26,97]. A single hydrothermal modification generally reduces the hydration power and solubility of starch. The addition of esterification modifications (acetylation or hydroxypropylation) can increase the hydration ability of granules and starch solubility. This statement is supported by several studies, which reported that the combination of acetylation + ANN modification in waxy cereal and potato starches had a higher swelling power (SP) than native starch and ANN starch, while the combination with HMT could increase SP, but this combination decreased the solubility of starch [10].

The substitution of ester groups (acetyl/hydroxypropyl) can weaken the crystalline structure of granules so that water penetration into the granules becomes easier and SP increases [10,23]. Nonetheless, Sitanggang et al. [97] reported that the combination of acetylation + ANN modification in black bean and pinto bean starches reduced SP and starch solubility. This decrease was due to an increase in starch crystallinity due to ANN modification, so the hydration power of the granules decreased [28]. In addition, the substitution of the acetyl group in this dual-modified starch causes the starch to become more hydrophobic [86,93].

Egodage [10] reported that the combination of acetylation + ANN modification in waxy rice and waxy potato starches increased the percentage of transmittance, which indicated an increase in the clarity of the paste. Low-temperature storage conditions caused a decrease in the clarity of the paste caused by the retrogradation process. The decrease in clarity of acetylated and acetylated + ANN-modified starch paste was not as big as that of natural starch and ANN-modified starch. The substitution of the acetyl group prevented amylopectin molecules from aggregating in the paste so that the retrogradation tendency decreased [10,77]. Reducing the retrogradation tendency could reduce the level of paste syneresis. Abedi et al. [130] reported that the combination of acetylation + sonication modifications reduced the percentage of syneresis due to the substitution of acetyl groups, which prevented the retrogradation process. Thus, the combination of acetylation + hydrothermal modification can improve the functional characteristics of starch. Changes in the functional properties of starch depend on the dominant modification treatment. Acetylation modification tends to weaken the crystalline structure of starch, so if this modification is combined with hydrothermal-modified starch, it can weaken the perfect structure. The substitution of ester groups with low DS tends to cause hydrophilic starch, thereby increasing the hydration ability of the modified starch, but if the DS is too low, the esterification reaction does not change the characteristics of the starch [93,137].

### 4.2. Pasting Properties of Acetylated Modified Starch/Flour

Pasting properties are indicators that determine starch properties during processing, which affect the cooking quality and functionality. Pasting properties described as an amylographic curve can be used to determine the application of starch in food ingredients [138,139]. Starch with low viscosity is suitable for liquid-based foodstuffs, while starch with high viscosity is suitable for use as a thickening agent [140,141,142].

Based on the amylographic curve, starch is classified into four types, namely types A, B, C, and V [143,144]. Starch type A shows high swelling of starch granules followed by a rapid decrease in viscosity during cooking, commonly found in potato starch, cassava starch, and several types of cereals. Type B starch, which has almost the same curved shape as type A but has a lower viscosity, is usually found in cereals. Type C starch, which exhibits limited swelling of the granules, has no peak viscosity, and tends to be heat-stable, is commonly found in modified starches and legumes. Type V starch, showing limited swelling of the granules, is usually found in starch that interacts with alcohol or fatty acid [144,145]. The effect of acetylation modification and its dual modification on the pasting properties of starch/flour is presented in Table 4.

In general, acetylation modification causes a decrease in starch gelatinization temperature so that it facilitates product application in terms of energy saving and its utilization in products that are susceptible to heat [128]. Several studies reported that the modification of acetylation causes a decrease in gelatinization temperature in sweet potato starch [128]; potato starch [147]; yam starches [84,85]; cowpea, yellow pea, and chickpea starches [62]; pinto bean and black bean starches [129]; rice starch [88,91]; waxy cornstarch [61]; and wheat starch [92], as well as decreased setback viscosity in sweet potato starch [104,128]; potato starch [127]; sword bean starch [90]; yellow pea, and chickpea starches [62]; black bean and pinto bean starches [129]; and rice starch [88]. A decrease in SB indicates a decreased retrogradation tendency, making the starch more stable at low-temperature storage. Meanwhile, most studies report that acetylation causes an increase in viscosity followed by an increase in starch hydration ability [59,62,88,90,92].

Table 4 also shows that acetylation modification combined with hydrothermal treatment can improve the pasting properties of starch to cover the weaknesses in hydrothermal modification. Sitanggang et al. [97] reported that the combination of acetylation + ANN modification could increase PT and SB and reduce PV and BD. The addition of the esterification treatment (acetylation/hydroxypropylation) was able to weaken the crystalline matrix so that it had a higher viscosity and a lower gelatinization temperature when compared to the single hydrothermal treatment [23,148,149].

The substitution of acetyl groups causes weak intermolecular forces, which decreases the gelatinization temperature of starch acetate. This statement is supported by several researchers who reported a decrease in gelatinization temperature after the modification of acetylation, including sweet potato starch [128], potato starch [147], yam starches [84,85], cowpea, yellow pea, and chickpea starches [62], pinto bean and black bean starches [129], rice starch [88,91], waxy maize starches [61], and wheat starch [92]. A decrease in the pasting temperature of the acetylated starch indicates that a large amount of energy is not required for the starch to gelatinize. This can be used to save energy in cooking or processing using acetylated starch [128].

The substitution of the acetyl group weakens the intermolecular bonds between amylose and amylopectin, thereby facilitating the penetration of water into the amorphous region, which is followed by an increase in the ability to absorb water during gelatinization [59,85,88,90]. Increasing the hydration ability of starch granules during gelatinization can increase the viscosity of the paste. Several studies reported an increase in paste viscosity after the modification of acetylation in sweet potato starch [127], purple yam starch [84], sword bean starch [90], yellow pea cowpea, and chickpea starches [62], buckwheat starch [59], waxy maize starch [61], rice starch [88], and wheat starch [92]. However, the effect of acetylation modification on starch viscosity is not always the same. Several studies reported that acetylation can have a different effect on each type of starch. The difference in results obtained is influenced by several factors, including the distribution of amylopectin chains, amylose content, molecular arrangement of granules, and the degree of substitution of acetyl groups [62,84].

Acetylation modification can reduce the setback viscosity (SB), which indicates an increase in starch storage stability, especially at cold temperatures. This statement is reinforced by several studies which reported a decrease in SB after modification of acetylation [62,88,90,104,127,129]. Acetyl group substitution inhibits intermolecular interactions of starch, thereby minimizing the tendency of starch retrogradation [88,94,104].

The modification of acetylation with different materials and types of starch can produce different pasting properties as well. Acetylation with vinyl acetate and acetic anhydride in starch can cause a decrease in the gelatinization temperature due to the weakening of the intermolecular interactions of starch [25,150]. However, acetylation with vinyl acetate in yellow pea and chickpea starches changed the type of starch gelatinization from type C to type B. In contrast, acetylation with acetic anhydride did not change the gelatinization pattern. The peak viscosity of all types of starch increased after acetylation modification with vinyl acetate. In contrast, acetylation with acetic anhydride caused a decrease in the peak viscosity of cowpea and chickpea starches. Both types of reagents caused a decrease in the SB of cowpea starch but an increase in SB in yellow pea starch. Meanwhile, acetylation with either acetate anhydride or vinyl acetate did not change the SB chickpea starch. Thus, the modification of acetylation with different reagents and starch sources has a different effect on its pasting properties [62,84,127].

Hydrothermal modification can increase the starch gelatinization temperature followed by a decrease in peak, BD, and SB viscosity, while acetylation causes a decrease in PT and SB followed by an increase in PV and BD [59,151,152]. Changes in the pasting properties of hydrothermally modified starch are contradictory to acetylation modifications. Sitanggang et al. [97] reported that the combination of acetylation + ANN modification led to an increase in PT and SB, followed by a decrease in peak viscosity and BD. This indicates that ANN modification is dominant because the resulting characteristics resemble ANN starch. In other studies, the dual modification of retrogradation + acetylation [94] and sonication + acetylation [130] produced starch with characteristics like acetylated starch. Yu et al. [94] reported that acetylation of retrograded starch caused a greater decrease in viscosity. This may be due to the substitution of acetyl groups causing starch to be hydrophobic. Referring to Rahim et al. [93], the greater the substituted acetyl group, the more hydrophobic the starch. In addition, acetylation treatment can increase the PV and BD and decrease the pasting temperature (PT). This is due to the substitution of acetyl groups weakening the crystalline structure so that water penetration into the granules is easier [11,41]. Thus, the pasting properties of the dual-modified starch will lead to the dominant treatment.

### 4.3. Starch Granule Morphology of Acetylated Modified Starch/Flour

The shape and size of starch granules vary depending on the starch source. Most of the tuber starches have oval-shaped granules, but some have round, polygonal, and spherical ones, and some are irregular; the size varies from 1 to 110 µm depending on the type of starch [153]. For example, sweet potato starch has polygonal granules [154,155], round, hexagonal, and spherical with a size of 4–26 µm and a smooth surface without cracks [104].

Changes in the morphology of starch granules are commonly found in modified starches. Several studies have stated that acetylation modification causes granule damage (deformation, fusion, cracking, resulting in holes) [84,88,94]. However, some modifications, such as annealing, generally do not cause significant changes in granule morphology. The morphological characteristics of modified starch may vary in each sample and are influenced by several factors, including the source of starch, the type of modification, and the modification conditions (time, temperature, and reagents used). The morphological characteristics of starch granules can be analyzed in several ways, including light microscopy, scanning electron microscope (SEM), transmission electron microscope (TEM), atomic force microscope (AFM), and confocal laser scanning microscope (CLSM) [27,156,157]. Information on the morphological characteristics of acetylated and its dual modification of starch/flour can be seen in Table 5.

In general, acetylation can cause the fusion and aggregation of starch granules [73,77,88,147]. This is related to the substitution of acetyl groups, which causes the disintegration of the structure and the more porous nature of the granules [104,159]. Xu et al. [73] reported that the substitution of acetyl groups in high numbers caused granule fragmentation, where the starch granules melted to form a fiber-like structure.

In addition, several studies have stated that acetylation reactions can also cause the formation of holes or pores so that the surface of the granules becomes rough [89,90,158]. These changes may occur due to partial hydrolysis by acids and reactions with alkalis. Fornal et al. [158] reported that the formation of holes or pores could be associated with the gelatinization of the granule surface due to the neutralization reaction with alkali (NaOH) in the acetylation modification process. However, several studies stated that acetylation modification at low DS did not cause changes in granule morphology [10,59,83,84]. Examples of morphological changes in acetylated starch can be seen in Figure 4.

Figure 4 shows that the modification of acetylation can cause starch granules to deform to form small fragments. The substitution of acetyl groups into starch molecules weakens the inter-/intramolecular bonds of starch and causes starch granules to lose their integrity [88,159]. The integrity of the starch is weakened, and its structure becomes increasingly porous during the acetylation reaction; then, the starch granules experience fragmentation and aggregate with each other [15,159].

Changes in the morphology of starch granules due to acetylation modifications can have different results. Several factors affect the morphological diversity of acetylated starch, namely internal factors such as the content of amylose-amylopectin and external factors such as type of reactant, concentration of reactant, reaction time, temperature, type of alkali, and concentration of alkali. Morphological changes will increase with the greater substitution of acetyl groups [126,160]. This may be due to the low DS acetylation reaction only taking place in the amorphous area of the granule surface so that it does not cause changes in the granule structure, whereas, at high DS, acetylation reactions can take place in the internal structure of the granules which causes greater damage [84,159].

The combination of acetylation with other modifications (dual modifications), such as hydrothermal treatment, also produces a variety of granule morphology. Acetylation modifications generally cause granule aggregation and fusion, whereas hydrothermal modifications (HMT and ANN) are more likely to maintain their integrity [10,161,162]. Therefore, in the dual treatment, the resulting modification of the morphological characteristics depends on the dominant modified treatment. However, there are dual-modified treatments that give a synergistic effect, while others depend on the most dominant treatment. The dual modification treatment that is synergistic includes sonication-acetylation modification. The sonication modification facilitates the acetylation reaction so that the effectiveness of the acetylation reaction in starch granules increases [130].

The dual modification of ANN-acetylation can produce starch with morphological characteristics resembling acetylated starch. This indicates that the modification of esterification/acetylation has a dominant effect [97]. Egodage [10] reported that the use of 5% ANN-Acetylation treatment did not cause granule changes, but 10% ANN-Acetylation treatment caused morphological changes resembling acetylated modified starch. This was due to the 5% acetylation of substituted acetyl groups that are too low to change the morphology of the granules. Although there was a slight change in the acetylated and dual-modified starch granules, the sizes and shapes of the granules did not change significantly; this indicates that the integrity of the granules was maintained during modification [10]. Different results were reported by Yu et al. [94], who stated that the dual modification of acetylation-retrogradation in starch and sweet potato flour caused granule deformation where the granules undergo fusion and aggregation. This could lead to granule damage when the ANN modification is smaller than during the retrogradation modification, so the ANN-acetylated starch is more stable in maintaining its structure than the retrogradation-acetylated starch. The differences in the characteristics produced are influenced by several factors, including the type of modification, the type of starch and its structure, as well as the conditions and treatment of the modification [56,163].

### 4.4. Starch Crystallinity of Acetylated Modified Starch/Flour

The crystallinity of starch can be determined by observing the X-ray diffraction pattern. The X-ray diffraction pattern is related to the formation of semicrystalline regions during modification so that the amorphous and crystalline areas can be identified [164,165]. The basic principle of this test is the exposure of X-rays to the sample by scanning the diffraction area at an angle of 2θ from 4°. The diffractogram pattern will produce a series of diffraction peaks with varying relative intensities along a certain value (2θ). Amorphous and crystalline regions can be distinguished by making curves and linear lines. The curve is made by connecting each point of minimum intensity; the area above the curve is known as the crystalline region (α_c_). The linear line is made by connecting two intensity points at 4° and 37° (2θ); the area that lies between the curve and the linear line is known as the amorphous region (α_a_). The ratio of the upper area (αc) to the total diffracted area (α_c_ + α_a_) is known as the degree of crystallinity or relative crystallinity [165,166,167,168].

Based on the intensity peaks formed, the starch crystallinity is divided into three types, namely types A, B, and C. The type A diffraction pattern has a distinctive pattern with peaks of 23°, 18°, 17°, and 15° (2θ), commonly found in cereal starch (rice) [169,170], and sweet potatoes [153,154,155,171]. The type B starch diffraction pattern is characterized by a small peak at 5.6° (2θ) and double peaks at 24° and 22° (2θ), commonly found in fruits, tubers, and high amylose starch [172,173]. Type C starch is a combination of different type A and type B crystal structures and is further classified into type C_A_ (close to type A) and C_B_ (close to type B). Type C starch showed strong diffraction peaks at 17° and 23° (2θ) and some small peaks around 5.6° and 15° (2θ). C_A_-type starch showed a shoulder peak at 18° (2θ), while C_B_ type showed two shoulder peaks at 22° and 24° (2θ) [167]. Type C starch is commonly found in beans and sweet potatoes. Besides types A, B, and C, there is type V, which is formed due to the presence of amylose-fat complexes. Lopez-Rubio et al. [174] V-type crystals show diffraction peaks at points 7°, 13°, and 20° (2θ). Guo et al. [141], in their research, stated that sweet potato has a type C diffraction pattern. The acetylation modification process and its combination can affect starch crystallinity. Information regarding the effect of acetylation modification and its combination on various types of starch on crystallinity can be seen in Table 6.

The type of starch crystallinity due to modification treatment can change, as indicated by the change in diffraction intensity. The acetylation treatment conditions and the dual modifications applied can change the polymorphic properties of starch. In general, acetylation at low DS can weaken the diffraction intensity, but no change in the crystalline diffraction pattern was found. This is supported by several studies, which stated that starch acetate weakened diffraction intensity but still retained its type of crystallinity [10,61,62,73,77,83,84,85,86,175]. However, the results obtained in each study were not always the same. Shah et al. [89] reported that acetylation did not change the type of starch crystallinity but caused a decrease in the diffraction intensity peak and found an increase in the peak 2θ of 20°. This peak may reflect the formation of amylose complexes with other compounds, such as amylose-lipids. In comparison, Adebowale et al. [90] reported that acetylation modification changed the crystallinity type of sword bean starch from type B to type C, indicating that polymorph A began to form during the modification.

The modification treatment of both acetylation and dual modification can increase, decrease, or not change the relative crystallinity (RC) of starch. The rearrangement of the double helix structure in starch granules can increase starch crystallinity, while the decrease occurs due to the partial gelatinization of starch granules [28,176]. The modification of starch caused an increase in RC, as seen in studies of acetylation modification of potato and cassava starch [83,147] and on the dual ANN-acetylation modification in mung bean starch [97]. Meanwhile, a decrease in RC occurred in the modified acetylation of white and purple yam starch [84], oat starch [89], high amylose maize starch [73], Banggai yam starch [175], waxy maize starch [86], waxy barley, waxy corn, waxy potato, waxy rice starch [10], and corn starch [79]. RC reduction also occurred in dual modifications acetylation-ANN of waxy potato starch, waxy barley starch, waxy rice starch, and waxy corn starch [10]. However, no RC changes were found in the modified acetylation of peas starch [62], cocoyam starch [85], and waxy maize starch [61].

The effect of acetylation on crystallinity depends on the type of starch and the treatment conditions. Differences in starch crystallinity are affected by several internal components of starch, including the interaction of double helices in crystals, the arrangement of double helices in the crystalline area, and the number of crystalline areas, which is affected by amylopectin content and chain length, and crystal size [164,165]. The polymorphic type and crystallinity of starch are also strongly affected by the internal components of starch and external factors such as environmental conditions and the presence of fat, which can form amylo-lipid complexes [28,177,178].

The substitution of hydroxyl groups with acetyl groups can weaken the hydrogen bonds that connect between starch molecules, which then causes a decrease in starch crystallinity [159]. This statement is supported by several studies, which stated that acetylation modification caused a decrease in RC [10,73,79,84,89,175]. Nonetheless, the crystallinity index of cassava and potato starch increased after acetylation modification [83,147]. This was due to the weakening of the amorphous area of the granules followed by an increase in the crystalline area. Meanwhile, in several studies, acetylation with low DS only took place in the amorphous area of the granules, so it did not cause any changes in the crystalline area [61,62,85]. The effect of acetylation modification on the X-ray diffractogram profile can be seen in Figure 5.

Figure 5 shows that the acetylation process does not cause changes in the diffraction peaks. Acetylation reactions with a low degree of substitution tend to attack amorphous areas so that no significant changes are found in the crystalline structure and do not change the diffraction pattern [14,85,175]. In Figure 5, two types of peaks are observed, namely the B-type peaks at 5.6° (2θ) and A-type peaks at 17.0° and 23.0° (2θ). The acetylation modification did not significantly change the A-type polymorphs at 17.0° and 23.0° (2θ) because these A-type polymorphs have a strong crystalline structure that is difficult to penetrate by acetylating reagents. However, the longer acetylation was able to reduce the number of B-type polymorphs at 5.6° (2θ), which had a weak crystalline structure. This could be due to the longer acetylation increasing the substitution of acetyl groups and reducing the hydroxyl groups in the amylose and amylopectin molecular chains, thereby damaging the long-range order of double helices so that the intra- and intermolecular bonds of starch weakened and caused a decrease in crystallinity [14]. However, an increase in crystallinity was found after 50 min of acetylation. This may have been due to the acid residue left after modification. This residue can cause the degradation of starch amorphous areas [175]. Wang et al. [179] reported an increase in crystallinity after the modification of acid hydrolysis because acid tends to attack amorphous areas. Acetylation with high DS can leave more acid residues and cause damage to amorphous areas. Thus, DS on acetylation modification in starch and flour greatly affects the crystallinity of the resulting starch.

Acetylation modification can be combined with several other modifications, especially hydrothermal modification, to obtain the desired crystallinity characteristics. In general, the combination of acetylation and hydrothermal modification (HMT, ANN) did not cause a change in the crystalline diffraction pattern, but a change in the RC did occur. The combination of ANN + acetylation modification caused a decrease in crystallinity in waxy (rice, barley, corn, and potato) starches [10]. Nonetheless, the dual ANN + AS modification of mung bean starch caused an increase in crystallinity, but the crystallinity was lower when compared to the single ANN treatment. This indicated that acetylation modification could disrupt the crystal structure of ANN starch, which was already perfect. The dual modification treatment could also weaken the effect of changing one of the treatments, such as the ANN + acetylation modification in mung bean starch [97].

The modification of acetylation and the combination of ANN + acetylation did not cause any changes in the starch crystalline diffraction pattern. However, the modification of acetylation could cause a decrease in RC and increases in DS. This was due to the substitution of acetyl groups weakening the formation of intermolecular hydrogen bonds, causing a weakening of the crystalline structure [86]. The dual combination of ANN + acetylation modification could cause a greater decrease in RC than ANN modification. This decrease indicated that the effects of these two modifications are opposite to each other. Structural changes that occurred during ANN led to an increase in the mobility of the amylopectin chains in the amorphous lamellae and the movement of molecules in the crystalline region, facilitating the entry of acetyl groups into the granules. This increase in acetyl group substitution then caused greater damage [10,15].

### 4.5. Comparison of Acetylated Modified Starch/Flour with Other Modifications

The modification of acetylation in starch/flour has several advantages, including a relatively easy modification process that can produce starch/flour, which has a high swelling ability, clear starch paste, and good stability against retrogradation [14,98,100,102]. However, acetylation modification has several drawbacks because it requires a process to clean up chemical residues, which is quite expensive; the potential for waste is not environmentally friendly; and acetylated starch is unstable to thermal processes [15,16,17]. In addition, different characteristics may occur depending on the type of starch/flour and the processing conditions. The general comparison of acetylated modified starch/flour with other modifications can be seen in Table 7.

## 5. Conclusions and Future Research

The modification of acetylation in starch/flour generally causes the fusion of starch granules, increasing the granule hydration ability, solubility, paste viscosity, storage stability and decreasing the gelatinization temperature and retrogradation stability. Acetylation generally does not change the crystalline structure of starch because it takes place in the amorphous areas of the granules, and some of them reduce the degree of crystallinity. Changes in starch/flour characteristics due to acetylation are strongly affected by the degree of acetyl group substitution. Dual acetylation modification with hydrothermal treatments such as HMT/ANN or cross-linking can produce starch/flour with better cooking and storage stability than native starch or single-modified starch and broaden its application to various products.

The modification of acetylation in starch/flour is continuing to develop, especially to increase the efficiency of the modification process and the application of acetylated starch in various fields and products. The efficiency of the modification process is being improved, including via pre-treatment through the formation of porous starch, such as by ultrasonication, partial hydrolysis, or oxidation. Acetylated starch/flour was also developed through dual modifications, including cross-linking and hydrothermal treatments, to increase thermal stability. The application of acetylated starch has also been developed more broadly to produce starch nanoparticles, which can then be used for encapsulation, starch-based composite, biofilms, drug delivery systems, and other applications.

## Figures and Tables

**Figure 1 polymers-15-02990-f001:**
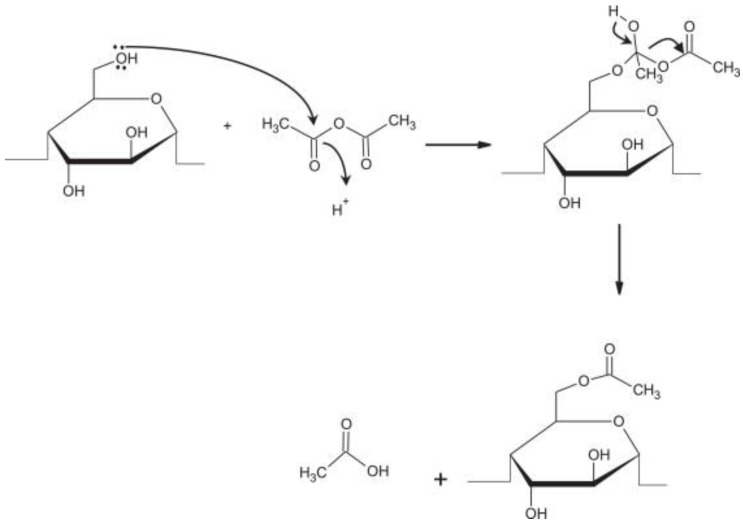
The basic principle of the acetylation reaction by the substitution of starch-free hydroxyl groups with acetyl groups [67], with permission from Elsevier, 2015.

**Figure 2 polymers-15-02990-f002:**
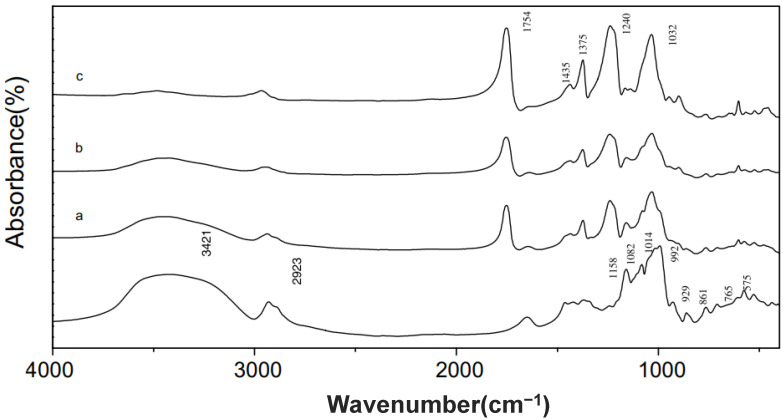
FTIR spectra of native corn starch and acetylated corn starches at different DS (a) 0.85, (b) 1.78, (c) 2.89 [100], with permission from Elsevier, 2008.

**Figure 3 polymers-15-02990-f003:**
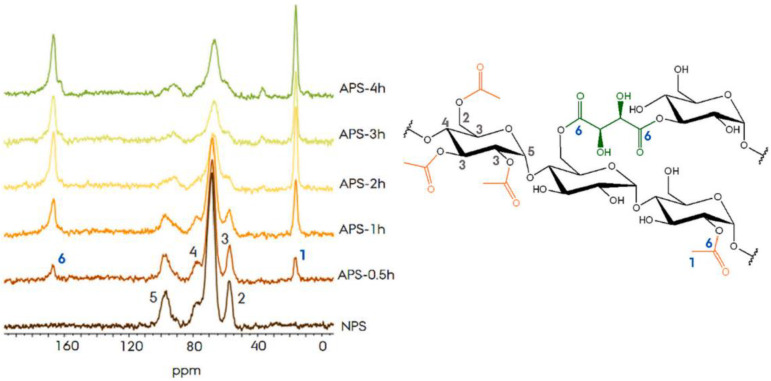
Single Pulse-^13^C NMR spectra of native pea starch (NPS) and acetylated pea starch (APS) by organocatalytic acetylation at different reaction times. Signal 1 corresponds to the carbon of the alkyl group, and signal 6 corresponds to the carbon of the ester groups of starch [101].

**Figure 4 polymers-15-02990-f004:**
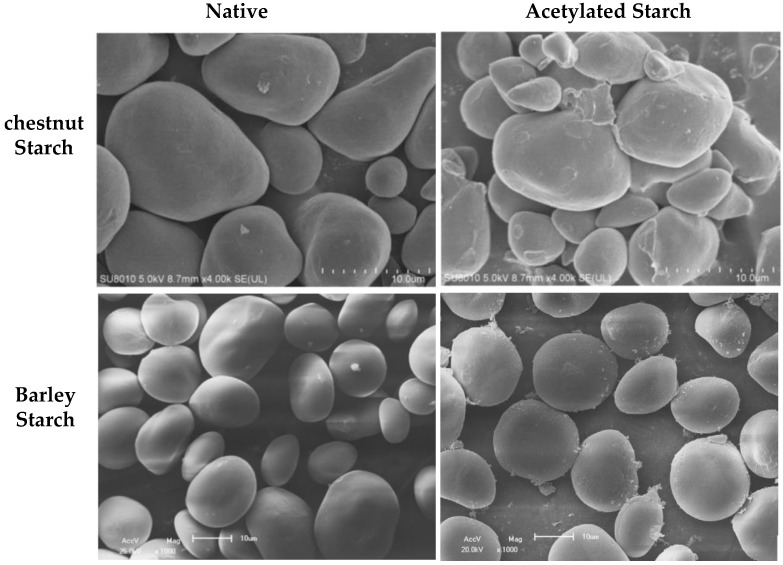
Morphological changes in acetylated modified chestnut starch [14] and barley starch [160], with permission from Elsevier, 2015.

**Figure 5 polymers-15-02990-f005:**
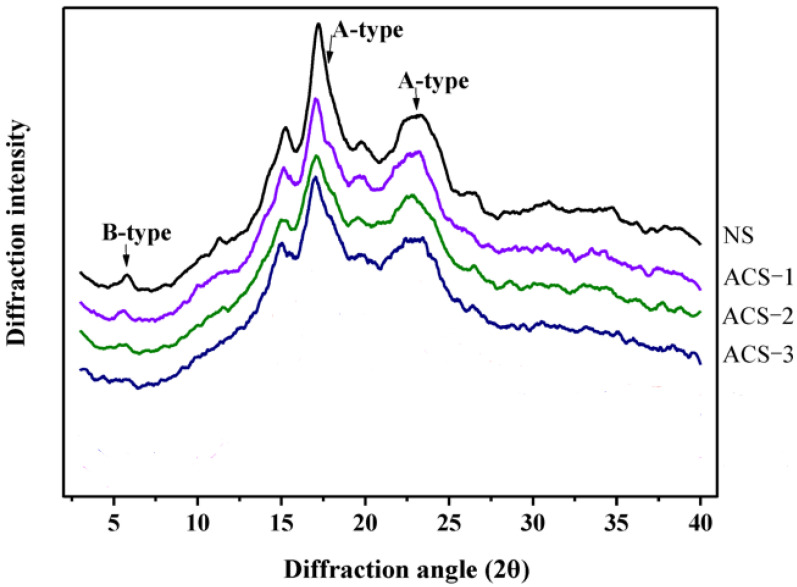
XRD profiles of acetylated modified chestnut starch at different reaction times (NS = native starch, ACS-1 = acetylated starch 30 min, ACS-2 = acetylated starch 60 min, ACS-3 = acetylated starch 90 min) [14].

**Table 1 polymers-15-02990-t001:** Some applications of acetylated modified starches/flours for various products and fields.

Starch/Flour and Treatment	Products/Applications	Characteristics	References
Acetylated cassava starch	Starch-based nanoparticles for encapsulation of antioxidants	Acetylated starch interacts well with antioxidant compounds, especially BHT, and protects antioxidants from degradation. Acetylated starch can increase the thermal stability of nanoparticles.	[30]
Acetylated rice starch nanocrystals	Nanocrystal for protein delivery	BSA protein release is significantly slowed. Acetylated rice starch nanocrystals can be good for protein delivery.	[31]
Acetylated debranched waxy corn starch	Nanocarrier for curcumin	Acetylated starch to be amphiphilic, having polar and non-polar groups. Curcumin micelles were spherical with a particle size of about 50–100 nm. Micelles of acetylated starch can accommodate curcumin until the concentration of 0.45 mg/mL	[32]
Retrograded acetylated corn starch	Drug (budesonide) delivery	Acetylation increased the hydrophobicity and reduced the swelling power and granule porosity. Tablets from retrograded acetylated corn starch released the drug in ileocolonic by 81.38%. Tablets were potentially suitable for the treatment of ileocolonic diseases.	[33]
Acetylated cassava starch and Maltol Incorporated	Active biodegradable film/packaging	The film based on acetylated cassava starch, which was incorporated with 10% maltol reduced molecular mobility and hydrophilicity; elongation was reduced by 34%, while the tensile strength was reduced by 37%. The active film inhibited the fungal growth for up to 6 times longer and maintained the flavor of bakery products.	[34]
Acetylated corn starch	Starch-based bioplastics	Acetylated corn starch improved the homogeneity and mechanical properties of biocomposites. The solubility of starch-based bioplastics decreased to 24.9–28.2%	[35]
Acetylated corn starch	Biodegradable polymers poly(lactic acid) for packaging materials	Acetylated corn starch increased the thermal stability of biodegradable polymers. Acetylated corn starch improved mechanical properties such as toughness and tensile strength.	[36]
Acetylated cassava starch	Starch nanoparticles for emulsion stabilizer	Acetylated starch nanoparticles with DS of 0.53 improved the hydrophobicity by a contact angle of more than 89.56°. Acetylated starch nanoparticles of 1.5% improved storage stability for up to 35 days and emulsion capacity by improving the droplet size and homogeneity.	[37]
Cross-linked acetylated cassava starch	Set yogurt	Cross-linked acetylated cassava starch improved the stability, viscous modulus (G″), and elastic modulus (G′) of the set yogurt.	[38]
Acetylated corn starch	Fat replacer of beef patties	The use of 15% acetylated corn starch improved the acceptance of organoleptic, microstructure, and physicochemical properties in beef patties, Acetylated corn starch is a suitable fat replacer for meat products.	[16]
Acetylated arenga starch	Bread	The addition of acetylated arenga starch up to 50% improved the quality of the bread produced, including sensorial properties, oven spring, oil absorption, and oil holding capacity.	[39]
Acetylated corn starch	Noodles	Acetylated corn starch increased the brightness of noodles. Acetylated starch reduced the tensile properties, chewiness, adhesion, and hardness of noodles. Acetylated starch increased the resistant starch and slowly digestible starch of noodles.	[40]
Acetylated rice starch and potato starch	Gut microbiota fermentation	Acetylated starch produced more SCFA than native starch. Acetylated starches were easier to use and more quickly fermentable by the gut microbiota.	[41]
Acetylated-crosslinked corn starch	Wood-based panel adhesive	The adhesive had better water resistance up to 1 MPa, The adhesive was also heat resistant, so it can be used in high-temperature pressing.	[42]
Acetylated waxy corn starch	Wood adhesive	The adhesive resistance to water increased up to 61.1% The shear strength increased up to 321% in the wet state and 59.4% in the dry state.	[43]
Acetylated corn starch and potato starch	Coagulants for wastewater treatment	Acetylated starch from corn and potato starch was effective as a coagulant for wastewater treatment by significantly reducing pH, color, turbidity, and electrical conductivity.	[44]

**Table 2 polymers-15-02990-t002:** Several methods of modifying acetylation on various types of starch/flour and their effect on DS values.

	Starches or Flours	Reagents/Acetylation Condition	Degree of Substitution (DS)	References
One Step Acetylation				
	Corn starch Waxy corn starch	Acetic anhydride, pH 8–9 using NaOH 2%.	Corn (0.05–0.07) Waxy corn (0.08–0.09)	[79]
	Maize starch	Choline carboxylate, imidazolium carboxylate, and imidazolium chloride	0.26–2.63	[80]
	Corn starch	Acetic anhydride using toluene sulfuric acid as the catalyst.	1.5 and 3.0	[81]
	Sweet potato starch Potato starch	Acetic anhydride	0.041–0.076	[82]
	Sweet potato starch	Acetic anhydride 0–8%, pH 8.1–8.3, NaOH 3%	0.032–0.123	[78]
	Potato starch Cassava starch	pH 8, NaOH 3%, 10–20 min	Potato starch (0.10–0.26) Cassava starch (0.10–0.18)	[83]
	Potato starch Corn starch	pH 8.0–8.4, NaOH 3%, 10 min	Potato (0.180–0.238) Corn (0.133–0.184)	[77]
	Purple yam (PY) starch White yam (WY) starch	pH 8.0–8.4, NaOH 3%, 10 min and 240 min	Purple yam (0.034–0.051) White yam (0.036–0.043)	[84]
	Cocoyam starch	pH 8.0–8.5, 1 M NaOH, 5 min	0.30	[85]
	Waxy maize starch	NaOH 20% for 40 min, and NaOH 2% for 120 min	0.12	[86]
	Waxy maize starch	pH 8.0–8.5, NaOH 1 N, KOH 1 N, and Ca(OH)_2_ 1 N for 60 min	0.077–0.085	[61]
	Maize starch	pH 8, 1 M NaOH, 60 min	0.080–0.210	[87]
	High amylose maize starch	pH 8, NaOH 50%, 15–240 min	0.57–2.23	[73]
	Rice starch	pH 8, NaOH 3%	0.03	[88]
	Oat starch	pH 8–8.5, NaOH 1 M for 5 min	0.02–0.05	[89]
	Sword bean starch	pH 8.0–8.4, 1 M NaOH, 30 min	0.14	[90]
	Yellow pea starch Chickpea starch Cowpea starch	Acetic anhydride, pH 7.5–9.0, 1–2 h, 20–25 °C, vinyl acetate, pH 9–10, 1–2 h, 20–25 °C	Acetic anhydride (0.059–0.066) Vinyl acetate (0.064–0.071)	[62]
	Japonica rice starch Indica rice starch	NaOH 4%, pH 7.8–8.2, 5 min	Japonica rice (0.066) Indica rice (0.060)	[91]
	Small granule wheat starch Large granule wheat starch	Acetic anhydride 8%, 30 °C, pH 8, NaOH 1 M, 15–20 min	0.039–0.043	[92]
	Sago starch	Acetylation (acetic anhydride, pH 7–10, NaOH 3%, T = room temperature, t = 50 min)	0.21–0.58	[93]
	Oat starch	Acetic anhydride (6% and 8%), pH 8.0–8.4, NaOH 3%, T = 25 °C, t = 10 min.	0.05–0.11	[21]
**Two steps acetylation**				
**Second step**	**Starches or flours**	**Reagents/Acetylation Condition**	**Degree of Substitution (DS)**	**References**
HMT	Buckwheat starch	Acetic anhydride, HMT= Moisture content 25%, temperature 110 °C, 4 h.	0.0289	[59]
Retrogradation	Purple sweet potato flour and starch	pH 8.5, NaOH 0,5 M, 15 min.	Purple sweet potato flour (0.08) Purple sweet potato starch (0.165)	[94]
Retrogradation	Potato starch	Acetic acid anhydride, pH 8–9, NaOH 3%, 15 min. Retrogradation using extruder, T = 60–65–70 °C, 100–110–120 °C or 150–160–170 °C.	3.1–4.4	[95]
Extrusion	Corn starch	Acetic anhydride 7.88%. Extrusion at Screw Speed (SS, 100 rpm) and Barrel Temperature (BT, 80 °C).	0.2	[96]
ANN	Mung bean starch	Acetic acid anhydride 20%, 15 min, pH 8 by NaOH 3%, 25 °C 25 min, ANN = 60 °C, 6 h.	0.02–0.26	[97]
ANN	Potato starch	Acetic acid anhydride, pH 8–9 by NaOH 0.5 M, 15 min, ANN = 51–61 °C, 48 h.	0.07–0.1	[26]
ANN	Waxy potato (WP) starch Waxy rice (WR) starch Waxy barley (WB) starch Waxy corn (WC) starch	Acetic anhydride 20 g/100 g, at pH 8, using NaOH 3%, 15 min, ANN = Moisture 75%, 10 °C below gelatinization temperature, 2–72 h.	WC (0.03–0.13) WB (0.06–0.24) WR (0.03–0.25) WP (0.01–0.12)	[10]

**Table 3 polymers-15-02990-t003:** The effect of acetylation and its dual modification on the functional properties of starch/flour.

Treatments	Materials	Functional Properties	References
Acetylation by vinyl acetate	Amaranth starch	Swelling power (SP) increased while solubility, WAC, and OAC decreased.	[126]
Acetylation	Sweet potato starch and flour Potato starch and flour	Starch: swelling volume (SV) increased; whiteness decreased. Flour: SV increased; whiteness increased. The whiteness degree of starch was higher than flour	[127]
Acetylation (pH 8.0–8.5, NaOH 1 M, 5 min)	Sweet potato starch	An increase in WAC, OAC, swelling power (SP), solubility, starch clarity, and gel strength. More stable to low-temperature storage.	[128]
Acetylation (Acetic anhydride 0–8%, pH 8.1–8.3, NaOH 3%)	Sweet potato starch	Swelling (SV) and solubility increased with increasing DS	[78]
Acetylation (pH 8, NaOH 3%, 10–20 menit)	Potato starch Cassava starch	Water binding capacity (WBC), paste clarity, and solubility increased Whiteness was decreased Acetylated potato starch, which was reacted for 20 min, caused a decrease in WBC.	[83]
Acetylation (pH 8.0–8.4, NaOH 3%, 10 and 240 min)	Purple yam (PY) starch White yam (WY) starch	In PY, there is an increase in WAC, SV, and solubility, but in 240 min of acetylated PY, the changes were not significant. At 240 min of acetylated WY, the increase in WAC and SV was greater, but the solubility value was lower than at 10 min of acetylated WY.	[84]
Acetylation (pH 8.0–8.5, 1 M NaOH, 5 min)	Cocoyam starch	SV increased with increasing temperature and pH Solubility was decreased Improved WAC, OAC, and paste clarity	[85]
Acetylation (pH 8.0–8.4, NaOH 3%, 10 min)	Potato starch Maize starch	SV and paste clarity increased Paste clarity potato starch > corn starch Solubility increased until the concentration of 8%, then decreased at 10–12%. WAC decreased	[77]
Acetylation (pH 8–8.5, 1 M NaOH, 5 min)	Oat starch	WAC and OAC increased Frozen storage stability increased	[89]
Acetylation (pH 8, NaOH 3%)	Rice starch	Aggregation and deformation of starch granules occurred Size of starch granules reduced Granules become perforated	[88]
Acetylation (pH 8.0–8.4, 1 M NaOH, 30 min)	Sword bean starch	WAC and SV increased Solubility decreased	[90]
Acetylation (acetic anhydride, pH 7.5–9.0, 20–25 °C, 1–2 h,) and (vinyl acetate, pH 9–10, 20–25 °C, 1–2 h)	Yellow pea starch Chickpea starch Cowpea starch	Acetylation by vinyl acetate had a greater swelling ability Acetylation by acetic anhydride in cowpea reduced the swelling ability	[62]
Acetylation (acetic anhydride, pH 7–10, NaOH 3%, T = room temperature, t = 50 min)	Sago starch	There was an increase in solubility, SP, oil absorption capacity, water absorption capacity, and clarity.	[93]
Acetylation and ANN	Black bean starch Pinto bean starch	Acetylation: SV increased ANN: SV decreased	[129]
Acetylation and HMT	Buckwheat seed starch	Acetylation: OAC, WAC, solubility, SV, paste clarity, and whiteness increased. HMT: WAC, OAC, and paste clarity increased, while solubility, SV, and whiteness decreased	[59]
Acetylation (8% Acetic anhydride, 30 °C, pH 8, 1 M NaOH, 15–20 min)	Small-sized granule wheat starch Large-sized granule wheat starch	SV and paste clarity increased, while FTS decreased SV in large granules was greater than in small granules	[92]
Acetylation (4% NaOH, pH 7.8–8.2, 5 min)	Japonica rice starch Indica rice starch	SV and solubility deceased	[91]
Acetylation (Acetic anhydride 6% and 8%, pH 8.0–8.4, NaOH 3%, T = 25°C, t = 10 min.)	Oat starch	Swelling factor increased Swelling temperature decreased Synaeresis decreased	[21]
Sonication-Acetylation (25, 40, and 25 + 40 Hz, 5 min, 45–75 °C)	Wheat starch	WAC and solubility increased frozen storage stability increased	[130]
Acetylation-ANN	Potato starch	Swelling power (SP) was higher than native starch and annealed starch	[26]
Acetylation-ANN	Mung beans starch	SV and solubility decreased	[97]
Acetylation and Acetylation-ANN	Waxy potato (WP) starch Waxy rice (WR) starch Waxy barley (WB) starch Waxy corn (WC) starch	Acetylation caused an increase in SV. Dual modification caused an increase in the SV of cereal starch, but there was no significant change in WP starch. Acetylation and Dual modified stable to low-temperature storage and paste clarity increased.	[10]
Acetylation-retrogradation (pH 8.5; NaOH 0.5 M, 15 min)	Purple sweet potato flour and starch	Solubility increased WAC and SP decreased	[94]
Acetylation-retrogradation (pH 8–9; NaOH 3%, 15 min)	Potato starch	Solubility increased water absorbability increased	[95]
Acetylation-Extrusion (Acetic anhydride 7.88%, Extrusion at BT = 80 °C, SS = 100 rpm)	Corn starch	Water absorption index decreased Water solubility index decreased	[96]

**Table 4 polymers-15-02990-t004:** The effect of acetylation and its dual modification on the pasting properties of starch/flour.

Treatments	Starch/Flour	Pasting Properties	References
Acetylation (pH 8.0–8.5, NaOH 1 M, 5 min)	Sweet potato starch	Peak viscosity (PV), setback viscosity (SB), breakdown viscosity (BD), and pasting temperature (PT) decreased.	[128]
Acetylation by vinyl acetate	Amaranth starch	PV and BD increased, while PT and final viscosity (FV) decreased.	[126]
Acetylation-Enzymatic	Sweet potato flour Potato flour	PV, BD, and SB decreased.	[104]
Acetylation	Sweet potato starch Potato starch	PT and SB of potato starch did not change significantly, but PV and BD decreased. Whereas in sweet potato starch, there was a decrease in PT, PV, BD, and SB. There was a decreased PT, PV, BD, and SB in acetylated flour.	[127]
Acetylation	Commercial potato starch	PT, PV, BD, SB, and final viscosity (FV) decreased.	[146]
Acetylation	Potato starch	PT, PV, and SB decreased, while BD increased.	[147]
Acetylation (pH 8.0–8.4, NaOH 3%, 10 and 240 min)	Purple yam (PY) starch White yam (WY) starch	The initial gelatinization temperature decreased, but heating to 95 °C increased the viscosity. PV and SB in purple yam increased, while PV and BD in white yam decreased.	[84]
Acetylation (pH 8.0–8.5, NaOH 1 M, 5 min)	Cocoyam starch	PT, PV, hot paste viscosity (HPv), cold paste viscosity (CPv), and SB decreased. BD increased.	[85]
Acetylation, NaOH (pH 8.0–8.5, NaOH 1 N, KOH 1 N, Ca(OH)_2_ 1 N, 60 min)	Waxy maize starch	PV increased, while PT decreased	[61]
Acetylation and HMT	Buckwheat seed starch	PV, FV, trough viscosity (TV), and PT for acetylated starch increased, while BD and SB decreased. PT in HMT-starch increased, while PV, FV, TV, BD, and SB decreased.	[59]
Acetylation (pH 8, NaOH 3%)	Rice starch	PV and TV increased, while PT and SB decreased.	[88]
Acetylation and ANN	Black bean starch Pinto bean starch	Acetylation-ANN reduces PV, HPV, CPV, BD, and SB Acetylation lowers PT and faster gelatinization time ANN increases PT and longer gelatinization time	[129]
Acetylation (pH 8.0–8.4, NaOH 1 M, 30 min)	Sword bean starch	PV, BD, and PT increased, while SB decreased	[90]
Cross-linked Acetylation starch (Acetic anhydride, pH 8, and sodium trimetaphosphate 0.7–0.9%)	Maize starch	PV and SB increased, while PT and BD decreased	[60]
Acetylation (acetic anhydride, pH 7.5–9.0, 1–2 h, 20–25 °C) and (vinyl acetate, pH 9–10, 1–2 h, 20–25 °C)	Yellow pea starch Chickpea starch Cowpea starch	The smaller the particle size of the chickpea and yellow pea, the greater the viscosity of the paste, while the particle size of the cowpea does not affect the viscosity. Acetylation using vinyl acetate produces a paste with a greater viscosity.	[62]
Acetylation (8% Acetic anhydride, 30 °C, pH 8, NaOH 1 M, 15–20 min)	Small granule wheat starch Large granule wheat starch	Acetylation increased paste viscosity, but PT decreased	[92]
Acetylation (4% NaOH, pH 7.8–8.2, 5 min)	Japonica rice starch Indica rice starch	PV and PT decreased; the reduction was greater with the single acetylation modification. BD and SB increased.	[91]
Acetylation-sonication (25, 40, and 25 + 40 Hz, 5 min, 45–75 °C)	Wheat starch	PV and BD increased, while PT and SB decreased	[130]
Acetylation-ANN	Mung bean starch	PV, BD, HPv, and CPv decreased, while PT and SB increased	[97]
Acetylation-retrogradation (pH 8.5, NaOH 0.5 M, 15 min)	Purple sweet potato flour and starch	Gelatinization occurs more quickly PV, BD, TV, FV, and SB decreased drastically	[94]

**Table 5 polymers-15-02990-t005:** The effect of acetylation and its dual modification on the morphological characteristics of starch/flour.

Treatments	Starch/Flour	Granule Morphology	References
Acetylated-Enzymatic	Sweet potato flour Potato flour	Aggregation of starch granules was formed, and the surface became irregular/rough.	[104]
Acetylation by vinyl acetate	Amaranth starch	The surface of the granules was smooth, showing, and no significant changes occurred.	[126]
Acetylated (pH 8, NaOH 3%, 10–20 min)	Potato starch Cassava starch	No significant changes occurred Slight granule fusion occurred	[83]
Acetylation (acetic acid, pH 8.0–8.5 by NaOH 0.5 N)	Corn starch Potato starch	The surface of starch granules in corn and potato starch becomes rough due to breakdown and erosion.	[44]
Acetylation	Potato starch	Fragmentation of starch granules	[147]
Acetylated (pH 8.0–8.4, NaOH 3%, 10 min)	Potato starch Corn starch	Fusion of starch granules occurred where potato starch granules were more susceptible than corn starch. The greater the concentration of reactants and DS, the greater the damage that occurred.	[77]
Acetylated (acetic anhydride) and acetylated distarch adipate (acetic anhydride and adipic acid)	Potato starch	Hole formation occurred in the middle of the granules (doughnut-like forms) in acetylated starch. There was no significant change, but the modified acetylated distarch adipate starch had a more compact surface.	[158]
Acetylated (pH 8.0–8.4, NaOH 3%, 10 and 240 min)	Purple yam (PY) starch White yam (WY) starch	Holes appeared, but starch granules tended to retain their shape	[84]
Acetylated (pH 8, NaOH 1 M, 60 min)	Corn starch	Granule aggregation occurred; the aggregation between granules was getting bigger along with the greater concentration of acetic anhydride.	[87]
Acetylated (pH 8, NaOH 50%, 15–240 min)	High-amylose maize starch	In the largest DS (2.23), granule fusion occurred, which caused the surface to become rougher and shaped like fibers. Granular damage was getting bigger along with the bigger the DS value.	[73]
Acetylated (pH 8, NaOH 3%)	Rice starch	Aggregation and deformation of starch granules occurred The size of the starch granules became smaller Starch granules became hollow	[88]
Acetylated (pH 8–8.5, NaOH 1 M, 5 min)	Oat starch	The texture of the starch granules becomes coarser, and small pores appear on the surface.	[89]
Acetylated (pH 8.0–8.4, NaOH 1 M, 30 min)	Sword bean starch	Some of the starch granules were broken (<10%) and caused the surface texture to become rough.	[90]
Acetylated (8% Acetic anhydride, 30 °C, pH 8, NaOH 1 M, 15–20 min)	Small granule wheat starch Large granule wheat starch	Starch granule damaged Starch with a larger granule size was easily damaged	[92]
Acetylation (Acetic anhydride 6% and 8%, pH 8.0–8.4, NaOH 3%, T = 25 °C, t = 10 min.)	Oat starch	The whole surface of starch granules was slightly damaged.	[21]
Acetylated and HMT	Buckwheat seed starch	The granules decreased slightly in size and became more separated from each other. Combination with HMT caused some starch granules to perforate.	[59]
Cross-linked Acetylation starch (Acetic anhydride, pH 8, and sodium trimetaphosphate 0.7–0.9%)	Maize starch	No significant changes occurred Starch granules have smooth surfaces and clear edges.	[60]
Acetylated and retrogradation (pH 8.5, NaOH 0.5 M, 15 min)	Purple sweet potato starch and flour	Acetylation caused granule aggregation and an increase in size Modification of acetylation + retrogradation resulted in a more compact structure, where the granule structure of starch is more compact than that of flour	[94]
Acetylated -sonication (25, 40, 65 Hz, 5 min, 45–75 °C)	Wheat starch	Starch granule fusion occurred There were holes and cracks on the surface of the starch granules	[130]
Acetylated and ANN	Mung bean starch	The granules became weaker and had a rougher surface The distance between granules became more tenuous, especially in dual-modified starch	[97]
Acetylated and ANN	Waxy potato (WP) starch Waxy rice (WR) starch Waxy barley (WB) starch Waxy corn (WC) starch	Acetylated starch had a rougher surface than native starch; the greater the damage, the greater the DS. The combination of acetylation + ANN caused a slight change in the granule surface in the form of a rougher surface.	[10]

**Table 6 polymers-15-02990-t006:** The effect of acetylation and its dual modification on the crystallinity of starch/flour.

Treatments	Starch/Flour	Crystallinity	References
Acetylation by acetic anhydride and toluene sulfuric acid	Corn starch	Acetylation reduced the degree of crystallinity of corn starch by increasing the amorphous regions. Acetylated starch with a DS of 3 had a slightly higher Tg than acetylated starch with a DS of 1.5.	[81]
Acetylation	Sweet potato Potato	There was no significant change in the crystalline region	[82]
Acetylation by vinyl acetate	Amaranth starch	There was no change in the crystalline diffraction pattern, but the relative crystallinity (RC) decreased.	[126]
Acetylation (pH 8, NaOH 3%, 10–20 min)	Potato starch Cassava starch	There was no change in the crystalline diffraction pattern, but the crystallinity index increased.	[83]
Acetylation	Potato starch	The degree of crystallinity increased	[147]
Acetylation (pH 8, NaOH 3%, 30–55 min)	Banggai yam starch	The crystalline diffraction pattern remained the same, but the degree of crystallinity increased. Acetylation for 50 min had the highest degree of crystallinity	[175]
Acetylation (pH 8.0–8.4, NaOH 3%, 10 and 240 min)	Purple yam (PY) starch White yam (WY) starch	The type of crystallinity remained the same, but the degree of crystallinity decreased with increasing DS.	[84]
Acetylation (pH 8.0–8.5, NaOH 1 M, 5 min)	Cocoyam	The crystalline diffraction pattern and degree of crystallinity did not change significantly.	[85]
Acetylation (NaOH 20%, 40 min, and NaOH 2%, 120 min)	Waxy maize starch	The crystalline diffraction pattern remained the same, but there was a decrease in the degree of crystallinity.	[86]
Acetylation (pH 8.0–8.5, NaOH 1 N, KOH 1 N, and Ca(OH)_2_ 1 N, 60 min)	Waxy maize starch	The crystalline diffraction pattern remained the same.	[61]
Acetylation (pH 8, NaOH 50%, 15–240 min)	High amylose maize starch	The crystalline diffraction pattern remained the same, but an increase in the degree of crystallinity.	[73]
Acetylation (pH 8–8.5, NaOH 1 M, 5 min)	Oat starch	The crystalline diffraction pattern remained the same, but there was a decrease in the degree of crystallinity.	[89]
Acetylation (pH 8.0–8.4, NaOH 1 M, 30 min)	Sword bean starch	There was a change in the type of starch crystallinity from type B to type C	[90]
Acetylation (acetic anhydride, pH 7.5–9.0, 1–2 h, 20–25 °C) and (vinyl acetate, pH 9–10, 1–2 h, 20–25 °C)	Yellow pea starch Chickpea starch Cowpea starch	The crystalline diffraction pattern and the degree of crystallinity of the starch did not change significantly.	[62]
Acetylation (acetic acid, pH 8.0–8.5 by NaOH 0.5 N)	Corn starch Potato starch	There was a similar diffraction pattern of starch crystallinity but a slight loss in the degree of starch crystallinity.	[44]
Acetylation	Corn starch Waxy corn starch	The crystalline diffraction pattern remained the same, but there was a decrease in the degree of starch crystallinity.	[79]
ANN-Acetylation	Mung bean starch	The crystalline diffraction pattern remained the same, The crystalline index increased, where the crystallinity of ANN was greater than the crystallinity of the dual modification.	[97]
ANN-Acetylation	Waxy corn (WC) starch Waxy barley (WB) starch Waxy rice (WR) starch Waxy potato (WP) starch	The degree of crystallinity decreased in acetylated starch, where the greater the DS, the greater the decrease. The greatest decrease in crystallinity occurred in the dual modification (ANN- Acetylation)	[10]
Sonication- Acetylation (25, 40, and 25 + 40 Hz, 5 min, 45–75 °C)	Wheat starch	There was a change in the type of starch crystallinity and a decrease in the degree of starch crystallinity as the sonication frequency increased in the dual modification (sonication-acetylation).	[130]

**Table 7 polymers-15-02990-t007:** The general comparison of acetylated modified starch/flour with other modifications.

Parameters	Acetylated Starch	Oxidized Starch	Crosslinked Starch	Hydrothermal Starch (HMT and Others)
Energy consumption for process modification	Low	Low	Low	High
Environmental friendliness	Less environmentally friendly	Less environmentally friendly	Less environmentally friendly	Environmentally friendly
Starch paste clarity	Paste clarity increases	Paste clarity increases	Paste clarity increases	Paste clarity decreases
Starch solubility	Solubility increases	Solubility increases	Solubility decreases	Solubility decreases
Swelling power (SP)	SP increases	SP increases	SP decreases	SP increases
Water absorption capacity (WAC)	WAC increases	WAC increases	WAC increases	WAC increases
Stability at high temperatures	The stability decreases	The stability increases	The stability increases	The stability increases
Stability against retrogradation	The stability increases	The stability increases	The stability decreases	The stability decreases
Degree of crystallinity	The degree of crystallinity decreases	The degree of crystallinity decreases	The degree of crystallinity increases	The degree of crystallinity increases
Starch granule morphology	The starch granules are rougher	The starch granules are rougher and more porous	The starch granules are rougher	The starch granules are rougher

## Data Availability

Not applicable.

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
