# Peer review of "Modification of Starches and Flours by Acetylation and Its Dual Modifications: A Review of Impact on Physicochemical Properties and Their Applications"

_polymers, 2023, doi:10.3390/polym15142990_

Round 1
Reviewer 1 Report
Subroto et al review the effects of acetylation modification and its dual modifications on the physicochemical properties of starch/flour and their applications. Acetylation can increase swelling power, swelling volume, water/oil absorption capacity, and retrogradation stability. Dual modification of acetylation with cross-linking or hydrothermal treatment can improve the thermal stability of starch/flour. The review is interesting. However, some points of the manuscript also should be improved.
1. The different molecular weight or other characters such as FTIR of starch/flour should be added in this review.
2. The nuclear magnetic resonance spectra after the acetylation modification of starch/flour should added in this review.
3. The authors should further discuss the solubility of starch/flour after acetylation modification.
4. The authors are encouraged to compare the acetylation modification with other methods.
Please carefully check the manuscript for writing and grammar.
Author Response
Response to Reviewer 1 Comments
Subroto et al review the effects of acetylation modification and its dual modifications on the physicochemical properties of starch/flour and their applications. Acetylation can increase swelling power, swelling volume, water/oil absorption capacity, and retrogradation stability. Dual modification of acetylation with cross-linking or hydrothermal treatment can improve the thermal stability of starch/flour. The review is interesting. However, some points of the manuscript also should be improved.
Point 1: The different molecular weight or other characters such as FTIR of starch/flour should be added in this review.
Response 1:
The different molecular weight or other characters such as FTIR of starch/flour have been added in the manuscript (Pages 10-11, Lines 247-266, red color).
Point 2: The nuclear magnetic resonance spectra after the acetylation modification of starch/flour should added in this review.
Response 2:
The nuclear magnetic resonance spectra after the acetylation modification of starch/flour have been added to the manuscript (Pages 11-12, lines 268-292, red color).
Point 3: The authors should further discuss the solubility of starch/flour after acetylation modification.
Response 3:
The solubility of starch/flour after acetylation modification has been further discussed in the manuscript (Page 16, lines 385-396, red color).
Point 4: The authors are encouraged to compare the acetylation modification with other methods.
Response 4:
The comparison of the acetylation modification with other methods has been added to the manuscript (Pages 30-31, lines 779-790, red color).

Reviewer 2 Report
Publish as it is.
The English language needs modification.
Author Response
Response to Reviewer 2 Comments
Publish as it is.
Response:
Thank you.

Reviewer 3 Report
Please, see attached file

-
Author Response
Response to Reviewer 3 Comments
Dear Authors
Your review is devoted to the analysis situation in the field of acetylation of polysugars, such as starches and flours. Information is interesting. Frankly speaking, I lacked examples proving one or another influence of some parameter on the properties of materials. Note that very often there was a repetition of the same information, but in different words.
Point 1: - In my opinion, text may be re-build, for example,
- Introduction
- Applications Acetylated Modified Starch/Flour
- Acetylation Modification Process in Starch/Flour
3.1. Mechanism of acetylation
3.2. Effect of method acetylation on properties of starch/flow
- Characteristics of Acetylated Modified Starch/Flour
4.1. Functional Properties of Acetylated Modified Starch/Flour
4.2. Pasting Properties of Acetylated Modified Starch/Flour
4.3. Starch Granule Morphology of Acetylated Modified Starch/Flour
4.4. Starch Crystallinity of Acetylated Modified Starch/Flour
Response 1:
Thank you for your opinion. The text has been re-built using this systematic.
- Introduction
- Applications Acetylated Modified Starch/Flour
- Acetylation Modification Process in Starch/Flour
3.1. Mechanism of acetylation
3.2. Effect of method acetylation on properties of starch/flow
- Characteristics of Acetylated Modified Starch/Flour
4.1. Functional Properties of Acetylated Modified Starch/Flour
4.2. Pasting Properties of Acetylated Modified Starch/Flour
4.3. Starch Granule Morphology of Acetylated Modified Starch/Flour
4.4. Starch Crystallinity of Acetylated Modified Starch/Flour
Point 2: The reaction mechanism should be considered in more detail (catalyst and solvent effect). Please, see References: (a) Advances in the Modification of Starch via Esterification for Enhanced Properties, Journal of Polymers and the Environment volume 29, pages1365–1379 (2021), https://link.springer.com/article/10.1007/s10924-020-02006-0 (b) Organocatalytic esterification of corn starches towards enhanced thermal stability and moisture resistance, Green. Chem., 2020, 22, 5017-5031 DOI: 10.1039/D0GC00681E.
Response 2:
The reaction mechanism has been considered in more detail (catalyst and solvent effect) using the References: (a) Advances in the Modification of Starch via Esterification for Enhanced Properties, Journal of Polymers and the Environment volume 29, pages1365–1379 (2021), https://link.springer.com/article/10.1007/s10924-020-02006-0 (b) Organocatalytic esterification of corn starches towards enhanced thermal stability and moisture resistance, Green. Chem., 2020, 22, 5017-5031 DOI: 10.1039/D0GC00681E. (Page 7, Lines 192-204, and page 35, Lines 987-992, red color).
Point 3: Table 1. I suggest to re-build this Table, for example,
Response 3:
Thank you for your suggestion. Table 1, which becomes Table 2, has been re-built according to the suggestion (Pages 8-10, Line 245-246, red color).
Point 4: Lines 119- 121. You wrote "Thus, the number of substituted acetyl groups is affected by several factors, including the source or type of starch, the concentration of the reactants, pH, reaction time, and the catalyst used [42,50]." In my opinion, it will be useful to add several examples for the proving this assertion.
Response 4:
Several examples for the proving assertion of "Thus, the number of substituted acetyl groups is affected by several factors, including the source or type of starch, the concentration of the reactants, pH, reaction time, and the catalyst used [42,50]." have been added (Page 7, Lines 225-240, red color).
Point 5: In Figure 3 two types of peaks are observed. It will be useful to explain these peaks
Response 5:
In Figure 3, which becomes Figure 5, The explanation of two types of peaks observed has been added in the text (Page 29, Line 740-749, red color).

Round 2
Reviewer 3 Report
-
-